

# CMTM6 significantly relates to PD-L1 and predicts the prognosis of gastric cancer patients

Xin Li[1,*], Ling Chen[2,*], Chuan Gu[3], Qiaoli Sun[4] and Jia Li[5]

[1] Department of Oncology, Longhua Hospital, Shanghai University of Traditional Chinese Medicine (TCM), Shanghai, China
[2] Department of Oncology, Yueyang Hospital of Integrated Traditional Chinese and Western Medicine, Shanghai University of Traditional Chinese Medicine (TCM), Shanghai, China
[3] Department of Plastic and Reconstructive Surgery, Shanghai Ninth People's Hospital, School of Medicine, Shanghai Jiaotong University, Shanghai, China
[4] Institute of Digestive Diseases, Longhua Hospital, Shanghai University of Traditional Chinese Medicine (TCM), Shanghai, China
[5] Department of Integrated Chinese and Western Medicine, Affiliated Cancer Hospital of Zhengzhou University and Henan Cancer Hospital, Zhengzhou, China
* These authors contributed equally to this work.

Corresponding authors
Qiaoli Sun, sunny5sunny@163.com
Jia Li, lijiacancerdoctor@126.com

## ABSTRACT

**Background:** The CKLF-like MARVEL transmembrane domain containing 6 (CMTM6) is a key regulator of the programed death receptor ligand-1 (PD-L1) protein. However, the usefulness of CMTM6 expression as a prognostic indicator and the relationship between CMTM6 and PD-L1 expression in gastric cancer (GC) remains unclear.

**Objectives:** We evaluated the expression and prognostic implications of CMTM6 in GC tissue and its relationship with PD-L1 expression.

**Patients and methods:** The protein expressions of CMTM6 and PD-L1 were detected in 122 cases of postoperative GC tissue using immunohistochemical (IHC) assays. Kaplan–Meier survival analysis was used to calculate the survival probability and a log-rank test was used to compare the survival curves. Univariate and multivariate Cox proportional hazard regression analyses were used to evaluate the clinically-related factors associated with survival. Pearson's correlation was used to determine the correlation analysis and estimate the statistical significance.
The univariate and multivariate logistic regression analyses were used to analyze the relationship between clinically-related factors and PD-L1 expression.

**Results:** Kaplan–Meier survival analysis showed that patients with high CMTM6 expression had shorter overall survival (OS) than those with low expression ($P < 0.001$). The expression of CMTM6 was an independent risk factor for prognosis in multivariate Cox proportional hazard regression analyses (HR:2.221, CI% [1.36–3.628], $P = 0.001$). The OS of patients with positively expressed PD-L1 was significantly shorter than those with negatively expressed PD-L1 ($P = 0.003$).
The expression of CMTM6 was significantly related to the positive expression of PD-L1 in gastric cancer tissues ($r = 0.186$, $P = 0.041$). The expression of CMTM6 was the independent risk factor for PD-L1 expression in multivariate logistic regression analysis (OR:2.538, CI% [1.128–5.714], $P = 0.024$).

**Conclusion:** CMTM6 expression is significantly related to PD-L1 and may be a useful prognostic indicator and a specific therapeutic target for cancer immunotherapy for GC patients.

## INTRODUCTION

Gastric cancer is one of the most common malignant cancers of the digestive tract in the world (*Globocan, 2018*). The majority of gastric cancer (GC) patients in China are diagnosed in the advanced stages of the disease (*Qiu et al., 2018*; *Liu et al., 2018*). The 5-year survival rate for patients with advanced gastric cancer who receive comprehensive surgery-based treatment is less than 30% (*Katai et al., 2018*). GC is commonly treated using chemotherapy and anti-HER-2 targeted therapy, which have limited efficacies (*Wagner et al., 2010*). Immune checkpoint inhibitors (ICIs) are relatively new in the treatment and diagnosis of many cancers but they have demonstrated unprecedented clinical efficacy in multiple types of cancer and have good clinical application prospects in gastric cancer (*Sharma & Allison, 2015*; *Fuchs et al., 2018*; *Shitara et al., 2018*; *Kang et al., 2017*). However, immunotherapy is only effective in a small proportion of gastric cancers and the expression of PD-L1 on tumor cells or tumor-infiltrating immune cells does not accurately predict for the patient's response to PD-1/PD-L1 inhibitors (*Braun, Burke & Van Allen, 2016*; *Guan et al., 2017*; *Wang & Wu, 2017*; *Pardoll, 2012*). Therefore, new immune-related therapeutic targets must be identified and their relationship with PD-L1 expression must be explored.

Two recent articles have identified CMTM6, which is a PD-L1 regulatory factor that plays an important role in inhibiting T-cell activation and anti-tumor responses. CMTM6 displays specificity for PD-L1 and promotes the expression of PD-L1 in tumor cells by T-cells. The depletion of CMTM6 decreases PD-L1 without compromising the cell surface expression of MHC Class I (*Burr et al., 2017*; *Mezzadra et al., 2017*). The role of CMTM6 in anti-tumor immunity and stabilization of the expression of PD-L1 has prompted us to explore its role in GC.

In this study, we evaluated the expression of CMTM6 and PD-L1 in 122 GC samples to investigate the prognostic success of CMTM6 and its relation to PD-L1 for GC patients.

## MATERIALS AND METHODS

### Tissue samples

We collected 122 tissue samples from GC patients who were surgically treated at the First Affiliated Hospital of Naval Military Medical University from December 2000 to June 2002. The study's follow-up deadline was October 31, 2007. Patients did not receive any radiation or chemotherapy before surgery. Tissue samples and clinical information were collected after patient consent was obtained and our study was approved by The Ethics

Committee of the First Affiliated Hospital of Naval Military Medical University
(ID: CHEC2014-098).

## Immunohistochemistry staining and results interpretation

All 122 gastric cancer samples were processed into tissue chips. The CMTM6 antibody was obtained from Novus Biologicals (NBP1-31183; Littleton, CO, USA) (*Zhu et al., 2019*). The PD-L1 antibody for IHC was purchased from Arigo Biolaboratories (ARG57681; Taiwan) (*Freeman et al., 2000*). The IHC kit was obtained from Maixin Biotechnologies (KIT-9730; Fuzhou, China) and was used according the manufacturer's instructions. The tissue chip was heated on a 63-degree baking sheet for 1 h. It was then dewaxed and hydrated in a fully automatic dyeing machine. The antigens were retrieved from the sample using a high-pressure heat repair method and the primary antibody was added (PD-L1 antibody 1: 50 and CMTM6 antibody 1: 200). The sample was returned to heat for 1 h at 37 °C and a goat anti-rabbit secondary antibody was added and heated for 30 min at 37 °C. 3,3-N-Diaminobenzidine Tertrahydrochloride (DAB) was developed for 1 min, hematoxylin was counterstained for 10 min, and the film was sealed.

Five fields from each specimen were randomly selected for analysis. The staining results were scored according to the Rahmen criteria and all of the IHC results were reviewed by two senior pathologists. Scores were used to assess CMTM6 expression according to the positive rate of staining: 0 points (negative), 1 point (1–25%), 2 points (26–50%), 3 points (51–75%) or 4 points (76–100%). X-tile plots were created by dividing CMTM6 expression into two populations: low- and high-level expression. All possible divisions of CMTM6 expression were assessed. Associations could be calculated at each division by log-rank test for survival. The data were represented graphically in a right-triangular grid where each point (pixel) represented the data from a given set of divisions. Data along the hypotenuse represented results from a single cut-point that divided the data into high or low subsets (*Camp, Dolled-Filhart & Rimm, 2004*). A total of 0.5 was the best cutoff value of CMTM6 expression for this set of samples. PD-L1 expression was considered to be positively expressed when the proportion of cells with membranous staining was >1% in neoplastic cells (*Calderaro et al., 2016*).

## Statistical analysis

The Kaplan–Meier method was used to calculate survival probability and the log-rank test was used to compare survival curves. The univariate and multivariate Cox proportional hazard regression analyses were used to evaluate the clinically-related factors associated with survival. The correlation analysis and estimated statistical significance was determined according to Pearson's correlation. The relationship between the clinically-related factors and PD-L1 expression were analyzed using univariate and multivariate logistic regression analyses. All statistical tests were two-sided, and $P < 0.05$ was considered to be statistically significant. SPSS 20.0 (SPSS, Chicago, IL, USA) was used for all statistical data analysis.

**Table 1 The clinicopathological characteristics of GC patients.**

| Variables | Number | % |
| --- | --- | --- |
| Age (year) | | |
| <60 | 52 | 42.6 |
| ≥60 | 70 | 57.4 |
| Sex | | |
| Male | 89 | 73.0 |
| Female | 33 | 27.0 |
| Tumor size (cm) | | |
| ≤3 | 26 | 21.3 |
| 3–6 | 68 | 55.7 |
| ≥6 | 28 | 23.0 |
| TNM stage (AJCC) | | |
| I | 42 | 34.4 |
| II | 48 | 39.4 |
| III | 32 | 26.2 |
| Grade (AJCC) | | |
| I + II | 80 | 65.6 |
| III | 42 | 34.4 |
| Histology | | |
| Adenocarcinoma | 107 | 87.7 |
| Others | 15 | 12.3 |
| CMTM6 | | |
| Low | 57 | 46.7 |
| High | 65 | 53.3 |
| PDL1 | | |
| Negative | 35 | 28.7 |
| Positive | 87 | 71.3 |

# RESULTS

## Clinicopathological characteristics of GC patients

The clinical characteristics of 122 gastric cancer patients are summarized in Table 1. A total of 52 (42.6%) patients were diagnosed at less than 60 years of age; 89 (73.0%) patients were male. A total of 26 (21.3%) patients had tumors ≤3 cm, 68 (55.7%) patients had tumors 3–6 cm and 28 (23.0%) patients had tumors >6 cm. A total of 42 patients (34.4%) had TNM stage I tumors, 48 patients (39.4%) had TNM stage II tumors and 32 patients (26.2%) had TNM stage III tumors. 80 patients (65.6%) had tumors with pathological grades I–II and 42 patients (34.4%) had grade III tumors according to the AJCC criteria staging system (*Liu et al., 2018*). The pathological morphology of patients with adenocarcinoma or a different tumor type was 107 (87.7%) and 15 (12.3%) respectively.

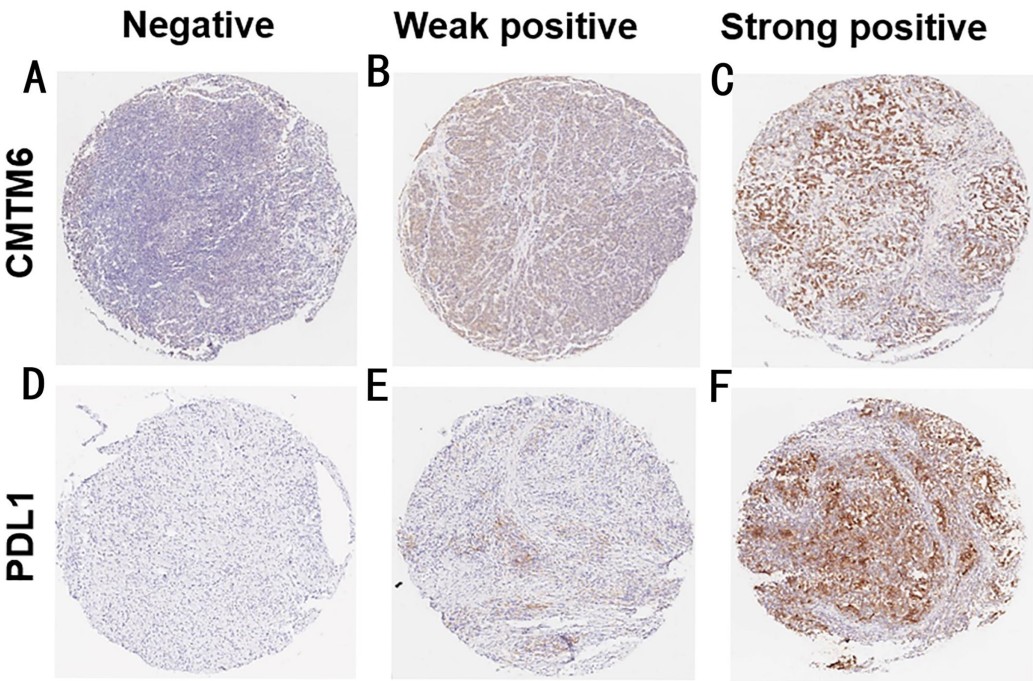

**Figure 1 The expression of CMTM6 and PD-L1 in GC tissues.** (A–C) Negative, weak positive and strong positive expression of CMTM6 (7×) (D–F) Negative, weak positive and strong positive expression of PD-L1 (7×).                 

## Expression of CMTM6 and PD-L1 in GC tissues

We detected the expression of CMTM6 and PD-L1in 122 gastric cancer tissues using an IHC assay. As shown in Fig. 1, CMTM6 is located mainly in the cell membrane and cytoplasm and the expression of PD-L1 was observed on the cell membrane. The low- and high-expressions of CMTM6 in gastric cancer tissues were 46.7% (57/122) and 53.3% (65/122), respectively, and the negative and positive expressions of PD-L1 were 35 (28.7%) and 87 (71.3%), respectively (Table 1).

## The prognostic value of CMTM6 expression in patients with GC

The cut-off level for CMTM6 expression was determined using X-tile software (Fig. 2A). Kaplan–Meier survival analysis showed that patients with high CMTM6 expression had shorter overall survival (OS) than these with low expression (Fig. 2B, $P < 0.001$). Univariate and multivariate Cox proportional hazard regression analyses were used to evaluate the clinically-related factors associated with survival. CMTM6 expression was an independent risk factor for prognosis (Table 2, HR:2.221, CI% [1.36–3.628], $P = 0.001$).

## Relationship between CMTM6 and PD-L1 in GC

Patients with positively expressed PD-L1 has a significantly shorter OS than those with negatively expressed PD-L1 (Fig. 3A) ($P = 0.003$). CMTM6 expression was significantly related to PD-L1 positivity in gastric cancer tissues (Fig. 3B) ($r = 0.186$, $P = 0.041$). Age, sex, tumor size, TNM stage, grade, histology, and CMTM6 expression were selected in the univariate and multivariate logistic regression analysis to further evaluate the

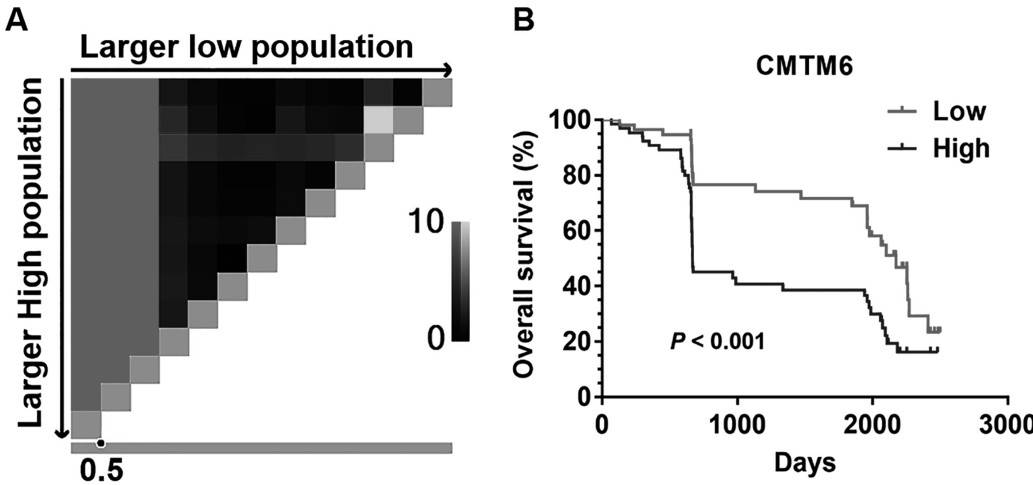

**Figure 2 The prognostic value of CMTM6 expression in patients with GC.** (A) X-tile plots Pattern diagram. The vertical axis represents all possible "high" populations, with the size of the high population increasing from top to bottom. The horizontal axis represents all possible "low" populations, with the size of the low population increasing from left to right. Coloration of the plot represents the strength of the association at each division, ranging from low (black) to high (gray or white). 0.5 is the best cutoff value of CMTM6 expression for this set of samples by X-title software. (B) Kapla–Meier analysis of overall survival (OS) with variable CMTM6 expression in GC patients. The black curve represents CMTM6 high expression group, and the gray curve represents CMTM6 low expression group.

relationship between the positive expression of PD-L1 and other clinically-related factors. The results showed that only CMTM6 expression was a risk factor of PD-L1 expression (Table 3) (OR:2.538, CI% [1.128–5.714], $P = 0.024$).

## DISCUSSION

CMTM6 belongs to a family of proteins made up of eight members (CMTM1-4 on the 16q21-22 chromosomal region, CMTM5 on the 14q11 region and CMTM6-8 on the 3p22 region) that play a role in epigenetic regulation, embryonic development and tumorigenesis (*Han et al., 2003*; *Sanchez-Pulido et al., 2002*; *Yafune et al., 2013*). CMTM6 is not required for PD-L1 maturation but co-localizes with PD-L1 at the plasma membrane and in recycling endosomes where it prevents PD-L1 from being targeted for lysosome-mediated degradation (*Burr et al., 2017*). CMTM6 was identified as a major regulator of PD-L1, a key immunological checkpoint, and a potential therapeutic target for tumor cell immune evasion.

There have been reports on CMTM6 for its regulation of PD-L1 and its function at the genetic and cellular levels for different cell lines. However, the usefulness of CMTM6 in clinical cancer samples has yet to be determined. We reported the prognostic value of CMTM6 and its relationship to PD-L1 in GC for the first time in our study. High CMTM6 expression was associated with shorter overall survival (OS) versus low expression ($P < 0.001$). Multivariate analysis confirmed the prognostic value (HR:2.221, CI% [1.36–3.628], $P = 0.001$). The expression of CMTM6 was significantly related to the positive presence of PD-L1 in gastric cancer tissues ($r = 0.186$, $P = 0.041$). The expression

Table 2 The univariate and multivariate Cox proportional hazard regression analyses between the clinical related factors and survival in GC patients.

| Variables | Univariate analysis | Multivariate analysis | |
|---|---|---|---|
| | Wald $\chi^2$ | HR (95%CI) | $P$ |
| Age (year) | 0.185 | | 0.667 |
| <60 | | Reference | |
| ≥60 | | 1.119 [0.672–1.862] | 0.667 |
| Sex | 0.206 | | 0.650 |
| Male | | Reference | |
| Female | | 0.881 [0.51–1.522] | 0.650 |
| Tumor size (cm) | 4.82 | | 0.090 |
| ≤3 | | Reference | |
| 3–6 | | 1.805 [0.954–3.414] | 0.069 |
| ≥6 | | 1.088 [0.493–2.402] | 0.834 |
| TNM stage (AJCC) | 0.514 | | 0.773 |
| I | | Reference | |
| II | | 0.887 [0.474–1.662] | 0.709 |
| III | | 1.136 [0.608–2.121] | 0.689 |
| Grade (AJCC) | 0.436 | | 0.509 |
| I + II | | Reference | |
| III | | 1.188 [0.712–1.984] | 0.509 |
| Histology | 2.599 | | 0.107 |
| Adenocarcinoma | | Reference | |
| Others | | 1.869 [0.874–3.997] | 0.107 |
| CMTM6 | 10.157 | | 0.001 |
| Low | | Reference | |
| High | | 2.221 [1.36–3.628] | 0.001 |
| PDL1 | 1.256 | | 0.262 |
| Negative | | Reference | |
| Positive | | 1.409 [0.774–2.566] | 0.262 |

of CMTM6 was the only independent risk factor of PD-L1 expression found by multivariate logistic regression analysis (OR:2.538, CI% [1.128–5.714], $P = 0.024$).

There are few studies reporting on CMTM6 expression and prognosis in other cancers. In primary pancreatic ductal adenocarcinoma (PDAC), a high CMTM6 expression was associated with shorter overall survival (OS). In the case of low expression, expression of CMTM6 also increased the prognostic value of PD-L1 expression (*Mamessier et al., 2018*). CMTM6 expression was associated with poor prognosis in hepatocellular carcinoma (HCC). The expression of CMTM6 was downregulated and was correlated with the AFP level and tumor metastasis of HCC patients (*Zhu et al., 2019*). CMTM6 overexpression was associated with poor prognosis in gliomas. CMTM6 is important for regulating T-cell activation and antitumor responses and is a promising target for developing immunotherapy of gliomas (*Guan et al., 2018*). A high CMTM6 expression was associated

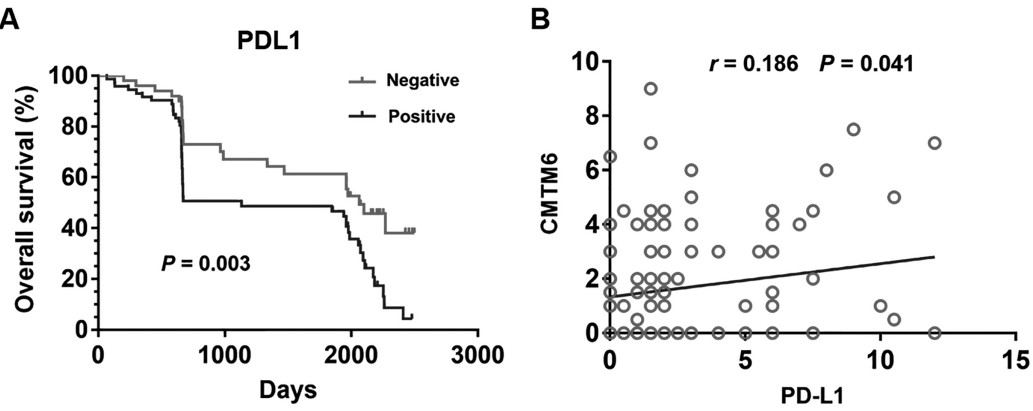

**Figure 3 Relationship between CMTM6 and PD-L1 in GC.** (A) Kapla–Meier analysis of overall survival (OS) with variable PD-L1 expression in GC patients. The black curve represents PD-L1 positive expression group, and the gray curve represents PD-L1 negative expression group. (B) The correlation of CMTM6 and PD-L1 expression in the GC tissues.

**Table 3 The univariate and multivariate logistic regression analysis between the clinical related risk factors and PD-L1 expression in GC patients.**

| Variables | Univariate analysis | Multivariate analysis | |
|---|---|---|---|
| | Wald $\chi^2$ | OR (95%CI) | *P* |
| Age (year) | 1.348 | | 0.246 |
| <60 | | Reference | |
| ≥60 | | 1.634 [0.713–3.741] | 0.246 |
| Sex | 0.572 | | 0.45 |
| Male | | Reference | |
| Female | | 1.473 [0.54–4.017] | 0.45 |
| Tumor size (cm) | 3.592 | | 0.166 |
| ≤3 | | Reference | |
| 3–6 | | 2.507 [0.946–6.645] | 0.065 |
| ≥6 | | 2.262 [0.695–7.358] | 0.175 |
| TNM stage (AJCC) | 2.719 | | 0.257 |
| I | | Reference | |
| II | | 0.561 [0.2–1.568] | 0.27 |
| III | | 1.392 [0.421–4.597] | 0.588 |
| Grade (AJCC) | 0.241 | | 0.624 |
| I + II | | Reference | |
| III | | 0.786 [0.3–2.059] | 0.624 |
| Histology | 0.044 | | 0.834 |
| Adenocarcinoma | | Reference | |
| Others | | 0.862 [0.215–3.464] | 0.834 |
| CMTM6 | 5.063 | | 0.024 |
| Low | | Reference | |
| High | | 2.538 [1.128–5.714] | 0.024 |

with reduced survival time and may be a strong indicator of poor prognosis in different cancer types.

CMTM6 maintains the stability of the PD-L1 cell surface expression and plays a key role in the survival of PD-L1 (*Mezzadra et al., 2017*). CMTM6 expression was positively correlated with PD-L1 in immunohistochemical and mRNA expression data of non-small cell lung cancer (*Koh et al., 2019*; *Gao et al., 2019*). Preventing CMTM6 from binding with PD-L1 may recover the existing immunosuppression response and serve as a promising target for immunotherapy. Zugazagoitia et al. found that high CMTM6 and PD-L1 co-expression in stromal cells, but not tumor cells, of lung cancer was significantly associated with longer overall survival in treated patients. However, this effect was not observed in the absence of immunotherapy (*Zugazagoitia et al., 2019*). *Koh et al. (2019)* also found that CMTM6 expression may be a promising tool for therapeutic decision-making regarding PD-1 inhibitors. *Zhao et al. (2019)* discovered a statistically significant interaction between CMTM6 and CD274 (PD-L1) indicating that CMTM4/6 may be a new therapeutic target for type I renal clear cell carcinoma (RCC) patients who are resistant to immune checkpoint blockade (ICB). CMTM6 expression was positively correlated with PD-L1 in GC, making it a promising immunotherapy target for GC.

This study also has certain limitations. Firstly, we used IHC to detect CMTM6 and PD-L1 expressions. However, this process is limited by positive criteria, the definition of cut-off values, and the specificity and reproducibility of the antibodies used. Secondly, due to some patients lacked the pathological data of Lauren classification, we did not perform a correlation analysis between CMTM6 and Laurent classification. Thirdly, because of the influence of various clinical factors, such as pathological type, tumor size, etc., TNM stage is not a risk factor affecting the prognosis of GC patients.

## CONCLUSION

CMTM6 expression is significantly related to PD-L1 and may be a useful prognostic indicator and a specific therapeutic target for cancer immunotherapy for GC patients.

### Funding

This work was supported by the Budget internal medicine research project of Shanghai Municipal Education Commission (2019LK003); the Longhua Hospital affiliated to Shanghai University of Traditional Chinese Medicine, Dragon Medical Scholars (Nursery Program) of National Clinical Research Base of Traditional Chinese Medicine (LYTD-82) and the Research project of Yueyang Hospital of Integrated Traditional Chinese and Western Medicine affiliated to Shanghai University of Traditional Chinese Medicine (2019YYQ25). The funders had no role in study design, data collection and analysis, decision to publish, or preparation of the manuscript.

## Grant Disclosures

The following grant information was disclosed by the authors:
Shanghai Municipal Education Commission: 2019LK003.
National Clinical Research Base of Traditional Chinese Medicine: LYTD-82.
Yueyang Hospital of Integrated Traditional Chinese.
Shanghai University of Traditional Chinese Medicine: 2019YYQ25.

## Competing Interests

The authors declare that they have no competing interests.

## Author Contributions

- Xin Li conceived and designed the experiments, performed the experiments, analyzed the data, prepared figures and/or tables, authored or reviewed drafts of the paper, and approved the final draft.
- Ling Chen performed the experiments, analyzed the data, prepared figures and/or tables, and approved the final draft.
- Chuan Gu analyzed the data, prepared figures and/or tables, and approved the final draft.
- Qiaoli Sun conceived and designed the experiments, performed the experiments, prepared figures and/or tables, authored or reviewed drafts of the paper, and approved the final draft.
- Jia Li conceived and designed the experiments, analyzed the data, prepared figures and/or tables, authored or reviewed drafts of the paper, and approved the final draft.

## Human Ethics

The following information was supplied relating to ethical approvals (i.e., approving body and any reference numbers):

The tissues and clinical information were obtained after patient consent. The research was approved by the Ethics Committee of the First Affiliated Hospital of Naval Military Medical University (ID: CHEC2014-098).

## Data Availability

Raw data are available as a Supplemental File.

## Supplemental Information

Supplemental information for this article can be found online at http://dx.doi.org/10.7717/peerj.9536#supplemental-information.

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
