# Peer review of "CMTM6 significantly relates to PD-L1 and predicts the prognosis of gastric cancer patients"

_PeerJ, doi:10.7717/peerj.9536_

## Round 0.1 · original submission · Major Revisions

Your manuscript has been reviewed by two experts in the field. As you can see from their comments below, both of them raise several points for its further revision. Please read their comments carefully and revise the manuscript accordingly. Particularly, please note that both of them mention about the cut-off value for the CMTM6 expression.

·

Basic reporting

.

Experimental design

.

Validity of the findings

.

Additional comments

1.The authors mentioned IHC score in the Method section. Was this score used for the assessment of CMTM6 expression?

2.The authors classified low and high expression of CMTM6. How did you classify the CMTM6 expression? The authors should showed the cut-off value of CMTM6 expression in the method section.

3. The authors mentioned that CMTM6 is co-localizes with PD-L1 at the plasma membrane in the Discussion section. However, the authors did not show this phenomenon. The author should showed the merge of immunofluorescence image of CMTM6 and PD-L1.

4. In the Discussion section, the authors mentioned that Recent research shows that CMTM6 displays132 specificity for PD-L1 and maintains its cell surface expression. CMTM6 depletion decreases PD-L1 without 133 compromising the cell surface expression of MHC class I and inhibits T cell activation and anti-tumor 134 responses in both in vitro and in vivo [15]. These sentences were already mentioned in the Introduction section. The authors should delete these sentences from the Discussion section.

5. In the Discussion section, the authors mentioned that the low expression of CMTM6 also increased the prognostic value of PD-L1 expression. This sentence give misunderstanding that low CMTM6 expression increased PD-L1 expression. The author should revise this sentence without misunderstanding.

Reviewer 2 ·

Basic reporting

no comment

Experimental design

1. Standardization of the new immunohistochemical protocol is very critical. It is better to use widely used antibodies for immunohistochemical study. Previous studies using Novus Biologicals (NBP1-31183, United States) and Biolaboratories (ARG57681, Taiwan, China) antibodies should be cited and positive controls also added.

Validity of the findings

1. Gastric cancer is mainly classified according to Lauren classification. The author should add Lauren classification and perform comparative analysis with CMTM6 expression.
2. In table2, the TNM stage does not affect prognosis. However, in general, the TNM stage is the most important prognostic factor, and it is difficult to understand that TNM tages I and III do not differ in prognosis. Therefore, the authors need to describe what selection bias they had when collecting samples.
3. The author should describe when the patient was operated and when the sample was taken.
4. In lane 115, the author said that the cut-off level for CMTM6 expression was determined using X-tile software. It is necessary to describe in detail how the cutoff is determined by X-tile software and accurately describe the cutoff value. A detailed explanation in Figure 2A is also needed.

---

## Round 0.2 · accepted · Accept

Your revised manuscript has been reviewed by one of the original reviewers and myself (unfortunately, the other original reviewer has declined this time). Both the accepted reviewer and I confirm that the revision has been appropriately done and thus I am happy to make the decision of its acceptance.

Reviewer 2 ·

Basic reporting

no comment

Experimental design

The manuscript was carefully revised according to my suggestion.

Validity of the findings

The manuscript was carefully revised according to my suggestion.

Additional comments

no comment